# RLPR: Extrapolating RLVR to general domains without verifiers

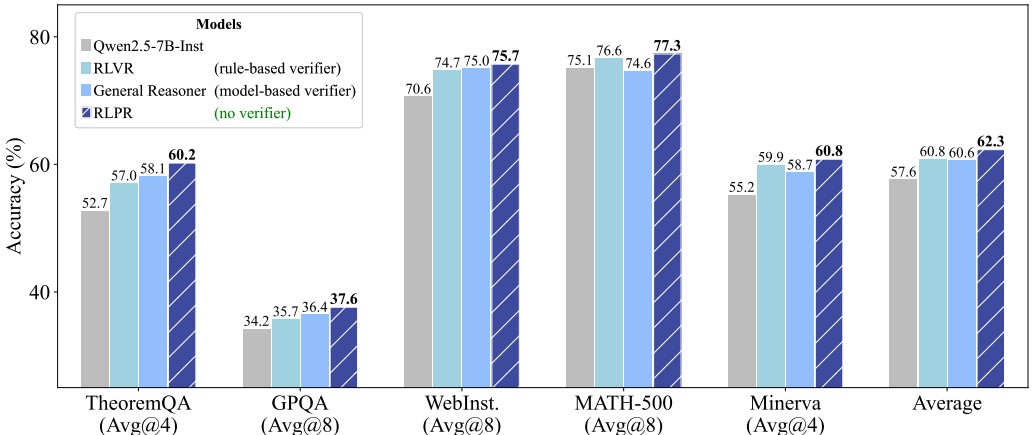

Figure 1: Overall performance on general-domain and mathematical reasoning benchmarks. By simply replacing the rule-based verifier reward of RLVR with the proposed LLM's intrinsic probability reward, **RLPR** achieves consistent improvements in both mathematical and general domains, even outperforming strong RL methods driven by model-based verifier reward. Average: average accuracy of five benchmarks. Verifier requirements of different methods are listed in parentheses.

## Abstract

Reinforcement Learning with Verifiable Rewards (RLVR) demonstrates promising potential in advancing the reasoning capabilities of LLMs. However, its success remains largely confined to mathematical and code domains. This primary limitation stems from the heavy reliance on domain-specific verifiers, which results in prohibitive complexity and limited scalability. To address the challenge, our key observation is that the LLM's intrinsic probability of generating a correct free-form answer directly indicates its own evaluation of the reasoning reward (i.e., how well the reasoning process leads to the correct answer). Building on this insight, we propose **RLPR**, a simple verifier-free framework that extrapolates RLVR to broader general domains. **RLPR** uses the LLM's own token probability scores for reference answers as the reward signal and maximizes the expected reward during training. We find that addressing the high variance of this noisy probability signal is crucial to make it work, and propose prob-to-reward and stabilizing methods to ensure a precise and stable reward from LLM intrinsic probabilities. Comprehensive experiments in four general-domain benchmarks and three mathematical benchmarks show that **RLPR** consistently improves reasoning capabilities in both areas for Gemma, Llama, and Qwen based models. Notably, **RLPR** outperforms concurrent VeriFree by 7.0 points on TheoremQA and 8.4 points on Minerva, and even surpasses strong verifier-model-dependent approaches General-Reasoner by 1.7 average points across seven benchmarks.

## 1 Introduction

Large-scale Reinforcement Learning with Verifiable Rewards (RLVR) has emerged as a promising paradigm to advance the reasoning capabilities of Large Language Models (LLMs) (Jaech et al.,

2024; DeepSeek-AI et al., 2025; Hu et al., 2025b; Luo et al., 2025a). This paradigm not only shows the power of scaling test-time computation for addressing complex problems, but also sheds valuable light on paths to AGI with incentivized exploration and evolution.

However, in contrast to pretraining that can learn foundational capabilities from general domain data, most RLVR methods are confined to mathematics (Hu et al., 2025b; Liu et al., 2025b; Zeng et al., 2025; Yu et al., 2025) and code generation (Luo et al., 2025a; He et al., 2025; Cui et al., 2025a). The primary reason is that existing RLVR methods heavily rely on domain-specific verifiers to obtain reward, as shown in Figure 2. The most widely adopted verifiers are handcrafted rules (Hu et al., 2025b; Liu et al., 2025b). Extending these rule-based reward systems to new models and domains typically requires prohibitive heuristic engineering. Moreover, for general-domain reasoning with free-form answers, it is even impossible to devise rule-based verifiers due to the high diversity and complexity of natural language. Recent works attempt to address this by training specialized LLMs as verifier models (Ma et al., 2025). However, training LLMs for general reward evaluation requires non-trivial and extensive data annotation, which often leads to unsatisfactory reward quality in practice. Involving separate verifier models also complicates RL frameworks and introduces additional computation cost. As a result, this scalability problem prevents existing RLVR methods from using rich general-domain data and limits the potential of broader reasoning capabilities.

To address the problem, we propose the **RLPR** framework (**R**einforcement **L**earning with Reference **P**robability **R**eward) that extrapolates general-domain RLVR without external verifiers. *The basic insight is that LLM's intrinsic probability of generating a correct answer directly indicates its own evaluation of the reasoning reward* (i.e., how well the reasoning process leads to the correct answer). It also reflects the policy by measuring how likely the LLM is to take the correct action. Therefore, we can directly leverage this probability signal as a reward to incentivize reasoning for the correct answer in general domains. Since this probability score is a natural built-in of LLM's foundational capabilities, it offers good coverage and potential for reward evaluation even without any specialized fine-tuning. It can also better deal with the complexity and diversity of free-form natural language answers, giving reasonable rewards even to partially correct answers.

Specifically, **RLPR** introduces two key innovations: (1) At the reward modeling level, we propose a simple and domain-agnostic prob-to-reward alternative to the explicit reward from external verifiers. We calculate the average decoding probabilities of the reference answer tokens as a Probability-based Reward (PR). Compared with naive sequence likelihood as reward (Zhou et al., 2025), the proposed PR shows better robustness and higher reward quality on quantitative examinations (see Figure 4). Moreover, we propose a simple debiasing method to eliminate the reward bias from text by optimizing the reward advantage over the same prompt without reasoning. (2) At the training level, we propose an adaptive curriculum learning mechanism to stabilize training. We adaptively remove prompts yielding low reward standard deviation (indicating prompts that are too easy or too complex), using a dynamic threshold based on the exponential moving average of past rewards' standard deviation. We find that this approach can well keep up with the reward distribution shifts during training, and improves both the training stability and final performance.

Comprehensive experiments on seven benchmarks show that, without any external verifiers, **RLPR** substantially enhances reasoning capabilities in both mathematical and general domains. Leveraging Qwen2.5-7B (Team, 2024) as base model, **RLPR** achieves 55.3 on MMLU-Pro and 60.2 on TheoremQA, even surpassing the strong General Reasoner-7B (Ma et al., 2025) that utilizes a specially trained 1.5B verifier model. Furthermore, compared with VeriFree (Zhou et al., 2025), a concurrent verifier-free approach, **RLPR** achieves significant improvement of 7.0 on TheoremQA and 8.4 on Minerva. We also evaluate **RLPR** on models from Llama3.1 (Grattafiori et al., 2024) and Gemma2 (Team et al., 2024), both achieving improvements of 5.3 overall points from training.

The contribution of this work can be summarized as fourfold: (1) We present **RLPR**, a simple and scalable framework that extends RLVR to general domains without using verifiers. (2) We propose a novel probability reward that eliminates the need for external verifiers and achieves better reward quality than naive likelihood as a reward. (3) We introduce a novel standard deviation filtering strategy that effectively stabilizes training by removing samples with low reward standard deviation. (4) We conduct comprehensive experiments to demonstrate the effectiveness of the proposed framework on various base models from Qwen, Llama and Gemma. All the codes, data, and model weights will be released to facilitate future research.

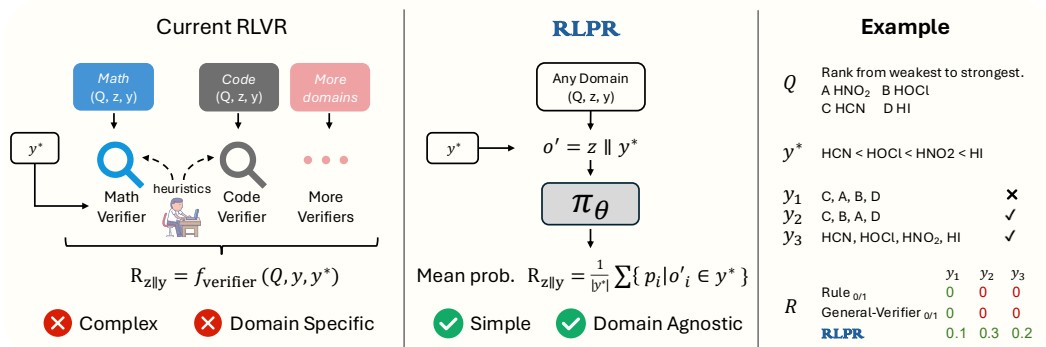

Figure 2: Existing RLVR methods rely on specialized verifiers for each domain, suffering from high complexity and limited scalability. We propose the **RLPR** framework, which replaces the complex verifier-based reward with a simple probability-based reward generated by the policy model $\pi_\theta$. $Q$: input question, $z$: generated reasoning content before final answer, $y$: generated final answer, $y^*$: reference answer. As shown in the example on the right side, rules and verifier models wrongly label both $y_2$ and $y_3$ as incorrect due to their limited capability of handling natural language complexity.

## 2 RLPR

In this section, we first introduce the fundamentals of RLVR and describe the procedure to calculate the probability reward for **RLPR**. Then we introduce the debiasing method and the standard deviation filtering approach.

### 2.1 REINFORCEMENT LEARNING FROM VERIFIABLE REWARDS

Reinforcement learning from verifiable reward (RLVR) is a general post-training paradigm in which a rule-based verifier assigns a scalar reward score to each generated response. Specifically, given a prompt $x$, the policy $\pi_\theta$ produces reasoning content $z$ and the final answer $y$. Then the expected verifier score is optimized:

$$\mathcal{J}(\theta) = \mathbb{E}_{z,y\sim\pi_\theta(\cdot|x)}\left[f_{\text{verifier}}(y, y^*)\right],\tag{1}$$

where $f_{\text{verifier}}$ is a task-specific, rule-based verifier checking whether the generated answer $y$ passes the test defined by ground truth $y^*$. Common instantiations include symbolic verifiers (Hynek & Greg, 2025) for mathematical problems or sandboxed execution (Bytedance-Seed-Foundation-Code-Team et al., 2025) for code generation. However, building rule-based verifiers is a laborious, systematic effort that involves designing handcrafted rules and edge case handling. This restricts the application of RLVR to new domains.

### 2.2 PROBABILITY REWARD

Motivated by the observation that the LLM's intrinsic probability of generating a correct answer directly indicates its internal evaluation of the reasoning quality, we use per-token decoding probabilities of the reference answer as the reward signal. Unlike existing methods that rely on domain-specific verifiers (Cui et al., 2025a; Luo et al., 2025a), which require substantial manual heuristics and engineering effort for the construction of verifiers, our reward computation process involves only the model itself. An overview of the process is illustrated in Figure 2.

We denote each response to question $Q$ with $o = (o_0, \cdots, o_N)$, where $o_i$ is an individual token in the response. To obtain probabilities, we first extract the generated answer $y$ from the full response sequence and denote the remaining content as reasoning $z$. We then construct a modified sequence $o' = (o'_0, \cdots, o'_{N'})$ by replacing the generated answer with the reference from the training data. This sequence is fed to the policy model to get probabilities $(p_0, \cdots, p_{N'})$. The probability reward is computed as:

$$r = f_{\text{seq}}(\{p_i|o'_i \in y^*\}),\tag{2}$$

where $f_{\text{seq}}$ aggregates per-token probabilities into a single reward scalar for the response $o$. While using $f_{\text{seq}} = \sqrt[n]{\prod}$ (the normalized product of probabilities, i.e., sequence likelihood) reflects the overall likelihood of the reference answer, we observe that it introduces high variance and is overly sensitive to minor variations, such as synonyms. For instance, the token probability sequences (0.01, 0.7, 0.9) and (0.05, 0.7, 0.9) yield vastly different scores under the product, despite only a small difference on the first token. To address this issue, we instead adopt $f_{\text{seq}} = \frac{1}{\lceil y^* \rceil} \sum$ (mean probabilities), which yields a more robust reward signal and demonstrates superior correlation with answer quality in our analyses (see Figure 4). We observe that probability reward values are highly consistent with the quality of the generated answer $y$, where high rewards are gained when the predicted answer is semantically similar to the reference answer and low rewards are assigned otherwise. Note that the length-normalization step is redundant for GRPO (Shao et al., 2024) but could be crucial for algorithms like REINFORCE++ (Hu et al., 2025a) which do not include group-normalization.

## 2.3 REWARD DEBIASING

Although the probability-based rewards correlate strongly with response quality, they are also influenced by various latent factors. We denote the contributors to the probability reward $r$ as $U_r$, which can be essentially decomposed into two latent factors:

$$U_r = U_z + U_{\text{others}}, \qquad (3)$$

where $U_z$ represents the effects of the reasoning content, and $U_{\text{others}}$ captures the characteristics of other related factors, such as text styles and word choices of the question and the reference answer. Using $r$ directly as a reward introduces bias associated with the unobserved factor $U_{\text{other}}$, potentially degrading the reward quality. To mitigate this, we introduce a base score $r'$ by computing the probability score of directly decoding the reference answer $y^*$, without intermediate reasoning $z$, using Eq 2. This gives $U_z = U_r - U_{r'}$, and the debiased probability reward is then calculated as:

$$\hat{r} = \text{clip}(0, 1, r - r'), \qquad (4)$$

where the clipping operation ensures that the reward remains within a favorable numeric range $[0, 1]$. This formulation effectively removes the potential bias from $U_Q$ and $U_{y^*}$ and models PR as the improvement in probability given the reasoning $z$. We observe that this debiasing step stabilizes training and enhances reward robustness. The final gradient estimator of our objective function is:

$$\begin{aligned} \nabla \mathcal{J}_{\text{RLPR}}(\theta) &= \nabla \mathbb{E}_{o \sim \pi_\theta(\cdot|x)} [\hat{r}] \\ &= \mathbb{E}_{o \sim \pi_\theta(\cdot|x)} [\hat{r} \nabla \log \pi_\theta(o|x)], \end{aligned} \qquad (5)$$

where we optimize the expected reward on the whole response $o = z || y$.

## 2.4 STANDARD DEVIATION FILTERING

Existing RLVR methods employ accuracy filtering (Cui et al., 2025a) to stabilize training by excluding too difficult and too easy prompts. Typically, this involves filtering entirely correct or incorrect prompts. However, the continuous nature of PR makes it challenging to directly apply accuracy filtering since it is hard to set a universal threshold for response correctness.

Through the analysis of accuracy filtering, we observe that filtering prompts with low standard deviation in reward values can effectively achieve a similar effect. Specifically, prompts that consistently yield all high or all low scores exhibit low standard deviation due to the boundedness of PR (i.e., all reward values lie within $[0, 1]$). Meanwhile, the overall standard deviation distribution continuously shifts during training, and a fixed threshold may cause either too strict or loose filtering at different training stages. To address this, we adopt an exponential moving average to dynamically update the filtering threshold $\beta$ using the average standard deviation of each training step. By filtering the prompts whose reward standard deviation is less than $\beta$, we introduce an adaptive curriculum learning mechanism to improve both the training stability and final performance.

## 3 EXPERIMENTS

In this section, we empirically investigate the effectiveness of **RLPR** in enhancing LLM reasoning capabilities. In addition to evaluating model performance, we also analyze the reward quality of our proposed PR, and the efficacy of different components.

### 3.1 EXPERIMENTAL SETUP

**Models.** We conduct experiments on Gemma2 (Team et al., 2024), Llama3.1 (Grattafiori et al., 2024) and Qwen2.5 (Team, 2024) series models for fair comparison with most existing methods and thorough evaluation. Unless otherwise specified, experiments are conducted on Qwen2.5-7B-Base.

**Training Data.** We adopt the collection of prompts released by (Ma et al., 2025), which includes high-quality reasoning questions across multiple complex domains. To focus on the effectiveness of **RLPR** in general domains, we only use non-mathematics prompts from the data. We ask GPT-4.1 (OpenAI, 2025) to filter out prompts that are too easy and finally get 77k prompts for training.

**Evaluation.** We evaluate the reasoning capabilities with multiple general reasoning and mathematical benchmarks. For math reasoning, we include MATH-500 (Cobbe et al., 2021), Minerva (Lewkowycz et al., 2022), and AIME24. For general domains, we adopt four benchmarks:

- MMLU-Pro (Wang et al., 2024) is a widely used multitask language understanding benchmark that includes challenging, reasoning-intensive questions across diverse domains. We randomly sample 1000 prompts from the benchmark to strike a balance between efficiency and variance.
- GPQA (Rein et al., 2023) includes graduate-level questions from multiple disciplines, including physics, chemistry, etc. We use the highest-quality GPQA-diamond subset for evaluation.
- TheoremQA (Chen et al., 2023) assesses a model's ability to apply theorems to solve complex science problems. This benchmark includes 800 high-quality questions covering 350 theorems from domains including Math, Physics, etc. We removed the 53 multimodal instructions.
- WebInstruct. We hold out a validation split from WebInstruct (Ma et al., 2025) as a more accessible benchmark for medium-sized models. Unlike the aforementioned benchmarks, this benchmark is designed to be less challenging while still assessing multidisciplinary reasoning. We uniformly sample 1k prompts from the training set and remove potential data contamination by applying 10-gram deduplication, resulting in 638 distinct questions.

**Baselines.** We compare our approach with the following established and contemporaneous methods: (1) **Base models and instruct models**. We include the Qwen2.5 (Team, 2024) model family for comparison, reporting results for both Qwen2.5-7B and Qwen2.5-7B-Instruct. We also compare with Gemma2-2B-it and Llama3.1-8B-Inst. (2) **PRIME** (Cui et al., 2025a) enhances the mathematical and code reasoning capabilities using implicit rewards. (3) **SimpleRL-Zoo** (Zeng et al., 2025) trains the model using rule-based rewards. We report both results of the Qwen2.5-Math and Qwen2.5-7B as the base model. (4) **Oat-Zero** (Liu et al., 2025b) proposes to remove the standard deviation and token-level normalization in GRPO. (5) **TTRL** (Zuo et al., 2025) eliminates the reliance on labeled reference answers and instead uses majority voting to assign pseudo-labels to sampled responses. We report the result of the model trained on MATH-500 (Zuo et al., 2025) prompts. (6) **General Reasoner** (Ma et al., 2025) conducts RLVR in multiple domains by introducing an additional verifier model, which is distilled from Gemini 2.0 (Google DeepMind, 2024) to verify general-domain responses. (7) **VeriFree** (Zhou et al., 2025) is a concurrent work that uses the likelihood of reference answers (for those shorter than 7-tokens) as the reward signal and incorporates an auxiliary fine-tuning loss. As results were only released for the Qwen3 (Team, 2025a) model series, we reproduce their methods on Qwen2.5-7B using the official repository. For fair comparison, we evaluate both their provided prompt and our training prompt, finding that the original prompt yields better results. Therefore, we adopt this configuration for this baseline.

**Implementation Details.** In each rollout step, we sample eight responses per prompt for a batch of 768 prompts using a temperature of 1, and subsequently perform 4 policy updates on the collected responses. The scale $\beta$ used for filtering is set to 0.5. The clip threshold in PPO loss is set to (0.8, 1.27) to prevent entropy collapse (Yu et al., 2025; Cui et al., 2025b). We evaluate multiple times on each benchmark with temperature of 1, reporting Avg@k and standard error of the mean (SEM). The max generation length for training and evaluation is 3072, with minimal truncation observed. For baseline evaluation, we follow the corresponding papers to select generation parameters and use our setup if the original paper uses greedy decoding. For reliable answer extraction, we adopt the R1 template (DeepSeek-AI et al., 2025) during training and use the striped content inside answer tags as the generated answer. For experiments on Gemma and Llama, we change the training and evaluation temperature to 0.6 and remove the <think> part in templates to prevent generation degradation. We observe that rule-based scoring scripts introduce errors in benchmarks containing question formats

| Model | Base | Verifier | MMLU-Pro Avg@8 | GPQA Avg@8 | TheoremQA Avg@4 | WebInst. Avg@8 | MATH-500 Avg@8 | Minerva Avg@4 | AIME 24 Avg@16 | General - | All - |
|---|---|---|---|---|---|---|---|---|---|---|---|
| | | | | | *Gemma Models* | | | | | | |
| Gemma2-2B-it | Base | – | $28.0_{\pm 0.2}$ | $20.6_{\pm 0.7}$ | $16.7_{\pm 0.5}$ | $33.4_{\pm 0.2}$ | $27.4_{\pm 0.4}$ | $15.8_{\pm 0.8}$ | $0.0_{\pm 0.0}$ | $24.7_{\pm 0.2}$ | $20.3_{\pm 0.2}$ |
| RLVR | Inst | Rule | $30.1_{\pm 0.3}$ | $\mathbf{27.0}_{\pm 0.5}$ | $21.3_{\pm 0.1}$ | $50.1_{\pm 0.3}$ | $29.4_{\pm 0.5}$ | $16.2_{\pm 0.6}$ | $\mathbf{0.6}_{\pm 0.3}$ | $32.1_{\pm 0.2}$ | $25.0_{\pm 0.2}$ |
| **RLPR** | Inst | ✗ | $\mathbf{31.8}_{\pm 0.2}$ | $25.5_{\pm 0.8}$ | $\mathbf{22.7}_{\pm 0.3}$ | $50.1_{\pm 0.3}$ | $\mathbf{30.8}_{\pm 0.3}$ | $\mathbf{18.2}_{\pm 0.5}$ | $0.2_{\pm 0.2}$ | $\mathbf{32.5}_{\pm 0.2}$ | $\mathbf{25.6}_{\pm 0.2}$ |
| | | | | | *Llama Models* | | | | | | |
| Llama3.1-8B-Inst | Base | – | $46.0_{\pm 0.4}$ | $27.7_{\pm 1.1}$ | $32.1_{\pm 0.7}$ | $53.8_{\pm 0.3}$ | $50.8_{\pm 0.5}$ | $34.9_{\pm 0.6}$ | $\mathbf{9.2}_{\pm 1.0}$ | $39.9_{\pm 0.3}$ | $36.4_{\pm 0.3}$ |
| RLVR | Inst | Rule | $48.3_{\pm 0.3}$ | $32.9_{\pm 0.7}$ | $34.1_{\pm 0.5}$ | $58.8_{\pm 0.3}$ | $49.7_{\pm 0.5}$ | $34.0_{\pm 0.7}$ | $3.8_{\pm 0.7}$ | $43.5_{\pm 0.3}$ | $37.4_{\pm 0.2}$ |
| **RLPR** | Inst | ✗ | $\mathbf{51.8}_{\pm 0.3}$ | $\mathbf{33.4}_{\pm 1.3}$ | $\mathbf{38.0}_{\pm 0.5}$ | $\mathbf{66.7}_{\pm 0.5}$ | $\mathbf{55.9}_{\pm 0.5}$ | $\mathbf{38.7}_{\pm 0.7}$ | $7.5_{\pm 1.1}$ | $\mathbf{47.5}_{\pm 0.4}$ | $\mathbf{41.7}_{\pm 0.3}$ |
| | | | | | *Qwen Models* | | | | | | |
| Qwen2.5-7B | – | – | $44.9_{\pm 1.3}$ | $29.6_{\pm 0.8}$ | $48.2_{\pm 1.0}$ | $63.6_{\pm 0.9}$ | $65.0_{\pm 1.1}$ | $41.2_{\pm 0.9}$ | $10.2_{\pm 1.3}$ | $46.6_{\pm 0.5}$ | $43.2_{\pm 0.4}$ |
| Qwen2.5-7B-Inst | Base | – | $55.0_{\pm 0.8}$ | $34.2_{\pm 0.7}$ | $52.7_{\pm 0.5}$ | $70.6_{\pm 1.0}$ | $75.1_{\pm 0.8}$ | $55.2_{\pm 0.5}$ | $13.1_{\pm 1.0}$ | $53.1_{\pm 0.4}$ | $50.8_{\pm 0.3}$ |
| Oat-Zero | Math | Rule | $46.5_{\pm 0.8}$ | $\mathbf{37.9}_{\pm 0.7}$ | $58.8_{\pm 0.6}$ | $72.2_{\pm 0.7}$ | $80.0_{\pm 0.7}$ | $54.3_{\pm 1.2}$ | $\mathbf{29.8}_{\pm 1.7}$ | $53.9_{\pm 0.4}$ | $54.2_{\pm 0.4}$ |
| PRIME | Math | Rule | $39.3_{\pm 0.6}$ | $31.3_{\pm 1.2}$ | $55.7_{\pm 0.5}$ | $58.4_{\pm 2.5}$ | $79.4_{\pm 1.0}$ | $48.7_{\pm 1.8}$ | $26.9_{\pm 2.1}$ | $46.2_{\pm 0.7}$ | $48.5_{\pm 0.6}$ |
| SimpleRL-Zoo | Math | Rule | $45.8_{\pm 0.7}$ | $36.6_{\pm 0.9}$ | $56.8_{\pm 0.8}$ | $70.8_{\pm 0.8}$ | $77.9_{\pm 0.9}$ | $55.4_{\pm 1.4}$ | $26.7_{\pm 2.4}$ | $52.5_{\pm 0.4}$ | $52.8_{\pm 0.5}$ |
| TTRL[†] | Base | Rule | $52.5_{\pm 0.6}$ | $33.5_{\pm 0.7}$ | $54.0_{\pm 0.2}$ | $70.6_{\pm 0.9}$ | $\mathbf{81.6}_{\pm 0.2}$ | $57.1_{\pm 0.6}$ | $14.8_{\pm 0.9}$ | $52.7_{\pm 0.3}$ | $52.0_{\pm 0.2}$ |
| SimpleRL-Zoo | Base | Rule | $54.1_{\pm 0.8}$ | $36.7_{\pm 0.7}$ | $53.9_{\pm 0.6}$ | $71.7_{\pm 0.9}$ | $76.5_{\pm 0.8}$ | $54.9_{\pm 0.6}$ | $16.3_{\pm 0.9}$ | $54.1_{\pm 0.4}$ | $52.0_{\pm 0.3}$ |
| RLVR | Base | Rule | $54.5_{\pm 0.6}$ | $35.7_{\pm 1.2}$ | $57.0_{\pm 0.5}$ | $74.7_{\pm 0.6}$ | $76.6_{\pm 1.2}$ | $59.9_{\pm 0.6}$ | $17.7_{\pm 1.2}$ | $55.5_{\pm 0.4}$ | $53.7_{\pm 0.3}$ |
| General Reasoner | Base | Model | $55.1_{\pm 1.0}$ | $36.4_{\pm 0.7}$ | $58.1_{\pm 1.0}$ | $75.0_{\pm 1.0}$ | $74.6_{\pm 0.7}$ | $58.7_{\pm 0.7}$ | $13.3_{\pm 1.4}$ | $56.1_{\pm 0.5}$ | $53.0_{\pm 0.4}$ |
| VeriFree[†] | Base | ✗ | $52.4_{\pm 0.6}$ | $34.9_{\pm 1.2}$ | $53.2_{\pm 0.5}$ | $69.5_{\pm 0.8}$ | $68.3_{\pm 1.4}$ | $52.4_{\pm 1.3}$ | $9.6_{\pm 1.3}$ | $52.5_{\pm 0.4}$ | $48.6_{\pm 0.4}$ |
| **RLPR** | Base | ✗ | $\mathbf{55.3}_{\pm 0.7}$ | $37.6_{\pm 0.9}$ | $\mathbf{60.2}_{\pm 0.3}$ | $\mathbf{75.7}_{\pm 0.6}$ | $77.3_{\pm 1.1}$ | $\mathbf{60.8}_{\pm 0.7}$ | $15.8_{\pm 1.3}$ | $\mathbf{57.2}_{\pm 0.3}$ | $\mathbf{54.7}_{\pm 0.3}$ |

Table 1: Overall performance on seven reasoning benchmarks. WebInst.: held-out evaluation set from WebInstruct. General: Average of MMLU-Pro, GPQA, TheoremQA and WebInst. [†]: Reproduced results, since no public Qwen2.5-based checkpoints are available.

beyond multiple-choice. To address this, we adopt GPT-4.1-mini for evaluation, and additionally leverage GPT-5 for more complex benchmarks, such as TheoremQA and Minerva. We set the k of TheoremQA and Minerva as 4 to save evaluation costs while maintaining a small SEM.

## 3.2 Main Results

The main experimental results are reported in Table 1, from which we observe that: (1) **RLPR** significantly improves general-domain reasoning performance. Without any external verifier, our method improves the average performance on four general-domain reasoning benchmarks by 22.7% on Qwen2.5-7B. (2) **RLPR** exceeds the RLVR baseline on Llama, Gemma and Qwen. Specifically, we achieve larger general reasoning performance improvement over RLVR for 4.0, 0.4 and 1.7 average points on Llama, Gemma and Qwen, respectively. (3) **RLPR** enhances mathematical reasoning capability on par with frameworks dedicated to math reasoning. Though we removed the mathematical category from the original WebInstrut dataset during training, we find the performance on multiple mathematical benchmarks is significantly improved and the score on Minerva surpasses Oat-Zero and SimpleRL-Zoo. (4) **RLPR** exhibits even better performance compared with methods that require trained verifier models, surpassing General Reasoner, which uses a trained 1.5B-parameter verifier model by 1.7 on average across all seven reasoning benchmarks. (5) **RLPR** achieves a significant performance advantage compared with concurrent verifier-free methods, with improvement of 7.0 points on TheoremQA and 8.4 points on Minerva over VeriFree (Zhou et al., 2025).

## 3.3 Probability-based Reward Analysis

We first illustrate a token-level probability example in Figure 3, where response sequence $o2$ receives a substantially lower score on the "HO" token, precisely reflecting the error made by response sequence $o2$ (i.e., placing option A before option B). For quantitative analysis of the Probability-based Reward (PR) quality, we sample eight responses for each prompt from the WebInstruct (Ma et al., 2025) and DeepScale (Luo et al., 2025b) datasets. To ensure a fair evaluation, we use the publicly released model from (Hu et al., 2025b). Three human annotators then evaluate the correctness of each response and we apply majority voting to ensure annotation quality. To control labeling costs, we randomly select 400 prompts from each dataset and stop annotation when 50 prompts containing both correct and incorrect responses are collected.

**PR discriminates correct responses better than the rule-based verifier on general data.** To evaluate the ability of different rewards to distinguish between correct and incorrect responses (i.e., assign higher rewards to correct responses), we rank responses for each prompt according to the respective rewards and compute the ROC-AUC (Bradley, 1997) metric using human annotations as ground truth. Higher AUC values indicate stronger discrimination capability. As shown in Figure 4, while the rule-based verifier achieves reasonable performance on mathematical prompts, it struggles on general-domain prompts, achieving an AUC of only 0.61. The primary flaw of the rule-based verifier in general domains is that it overlooks correct responses due to its limited capability of processing natural language complexity. We show an example in Figure 2 to illustrate the phenomenon. In contrast, PR consistently delivers high-quality rewards across both mathematical and general domains.

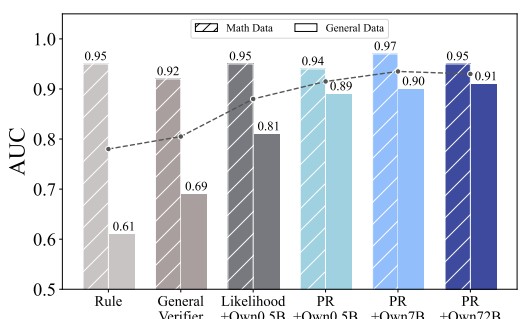

Figure 3: Token-level probability visualization, where deeper colors represent higher values. The underlined part highlights that probabilities precisely reflect that response sequence $o2$ incorrectly place option B after A, resulting lower scores at the corresponding positions in the reference answer. The question is shown in Figure 2.

**PR outperforms verifier models across both mathematical and general domains.** While the General-Verifier achieves improvement over rule-based reward on general data ($0.61 \rightarrow 0.69$), its performance declines on mathematical prompts ($0.95 \rightarrow 0.92$) as shown in Figure 4. We attribute this limitation to the finetuning-based paradigm, which requires extensive task-specific data and struggles to generalize across domains. In contrast, our proposed PR achieves improvements of at least 2% on mathematical data and 20% on general-domain data compared with the verifier model. Upon analyzing the General-Verifier's judgments, we find that its main errors stem from limited comprehension of complex responses and challenges in output parsing. By leveraging the intrinsic capabilities of LLMs, PR directly produces high-quality reward scores in a single forward pass, also eliminating the need for any text post-processing.

Figure 4: Reward quality comparison. We report the AUC on both math data and general data, and highlight the average score with the dashed line. Qwn: Qwen2.5 models.

**PR is effective with even small-scale models.** We compare the quality of PR using models of varying sizes. As shown in Figure 4, even the smallest Qwen2.5-0.5B outperforms the specifically trained General-Verifier on both mathematical and general data. While increasing the model size further improves the performance on general-domain data, gains on mathematical data are marginal due to the already high absolute scores.

**PR is robust over entropy and length distribution.** We also analyze the robustness of PR by analyzing the correlation between PR values and factors, including length and decoding entropy of generated responses. For each prompt, we calculate the Spearman correlation coefficient and p-value. We observe that only 8% prompts get a p-value smaller than 0.05, and the average coefficient is -0.060 for length and 0.059 for entropy. These results indicate that the probability reward values show negligible correlation with both entropy and length. This indicates that our proposed reward serves as a robust reward mechanism.

### 3.4 Ablation Study

To investigate the contribution of different design choices in **RLPR**, we perform an ablation study.

**Effect of per-token probability as reward.** We compare our per-token probability-based reward with naive sequence likelihood as the reward signal. In the calculation of likelihood, low-probability tokens can dramatically affect the final reward. For instance, probabilities of $1e^{-4}$ versus $1e^{-5}$ can lead to a tenfold difference in reward, despite their small absolute difference. This issue becomes

| Method | MMLU-Pro Avg@8 | GPQA Avg@8 | TheoremQA Avg@4 | WebInst. Avg@8 | MATH-500 Avg@8 | Minerva Avg@4 | AIME 24 Avg@16 | General - | All - |
|---|---|---|---|---|---|---|---|---|---|
| **RLPR** | **55.3** | **37.6** | **60.2** | **75.7** | 77.3 | **60.8** | **15.8** | **57.2** | **54.7** |
| w/o debiasing | 54.0-1.3 | 36.6-1.0 | 59.1-1.1 | 75.1-0.6 | **77.6**+0.3 | 60.2-0.6 | 13.8-2.0 | 56.2-1.0 | 53.7-1.0 |
| w/o std-filtering | 50.4-4.9 | 34.5-3.1 | 57.5-2.7 | 75.2-0.5 | 76.6-0.7 | 59.6-1.2 | 15.6-0.2 | 54.4-2.8 | 52.8-1.9 |
| w/o token prob. | 39.9-15.4 | 33.8-3.8 | 37.8-22.4 | 59.6-16.1 | 51.7-25.6 | 34.8-26.0 | 1.9-13.9 | 42.8-14.4 | 37.1-17.6 |

Table 2: Ablation experimental results. Token prob.: token probability average.

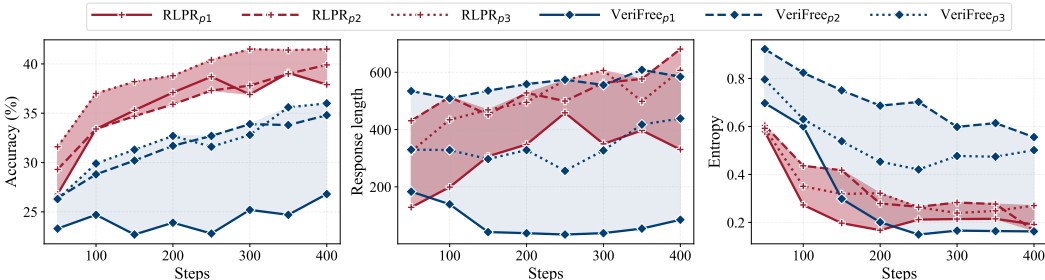

Figure 5: Robustness across different training prompt templates. **RLPR** yields consistently higher performance compared with VeriFree. Left: average performance on seven benchmarks. Middle: response length. Right: response entropy during training.

more pronounced for longer reference answers, which are more likely to contain at least one low-probability token. Zhou et al. (2025) address this instability by filtering out prompts whose reference answers exceed seven tokens. However, this also significantly limits the data diversity. In contrast, using the mean per-token probability is much more robust and yields better performance, as shown in Table 2. We also compare the reward quality of the likelihood reward and our proposed PR in Figure 4, where PR consistently achieves better results on both domains.

**Effect of reward debiasing and standard deviation filtering.** We compare our final debiased reward $\hat{r}$ with directly using the reward in Eq 2. Results in Table 2 shows that the performance on both benchmarks is worse with original reward, demonstrating the effectiveness of the debiasing operation. To quantify the effectiveness of the standard deviation filtering approach, we also train a model without any filtering mechanism. The results in Table 2 show that the filtering strategy is important for the final performance of models by removing prompts that do not get diverse responses.

### 3.5 ROBUSTNESS ANALYSIS

Compared with rule-based rewards, the distribution of our proposed probability-based reward may be influenced by variations in training prompt templates. To evaluate the robustness of **RLPR** with different templates, we consider three prompt settings: $p_1$ from VeriFree (Zhou et al., 2025), $p_2$ used in DeepSeek-R1 (DeepSeek-AI et al., 2025) and $p_3$ which places the format requirement in user prompt. To reduce training costs, we switch the base model to Qwen2.5-3B, decrease the batch size to 128, and apply a single update per training step. For fair comparison, we adopt the original dataset from VeriFree for this experiment. Figure 5 presents the comparison of performances, response length, and entropy across different training steps. We observe that **RLPR** maintains consistent performance regardless of prompt choice, while VeriFree exhibits high sensitivity, with a notable performance drop of by 8.0 at step-400 when using $p_1$. Furthermore, the response length of **RLPR** under all prompts converges to a similar level, and the entropy remains within a reasonable range with no signs of entropy collapse (Cui et al., 2025b).

### 3.6 COMPARISON WITH UNSUPERVISED RL

Recent unsupervised RL methods including TTRL (Zuo et al., 2025) and ScPO(Prasad et al., 2025) adopt frequency-based rewards that are conceptually similar to our probability-based reward. These approaches rely on external verifiers to assess the equivalence between answers and then group answers to get frequencies. More frequent answers get higher rewards. In this way, these approaches bypass the need for explicit labels. However, this reliance on external parsers makes it difficult to scale these methods to general domains that involve free-form answers. More importantly, these

methods only amplify the probability of generating the currently most likely response, regardless of its correctness. As a result, if the policy model is already prone to generating incorrect responses, these methods will reinforce this erroneous behavior. In contrast, **RLPR** assigns higher scores to responses that are more likely to lead to the correct answer, addressing this critical limitation.

## 4 RELATED WORKS

**Reinforcement Learning with Verifiable Rewards**. Reinforcement learning from binary verifiable rewards (Cui et al., 2025a; Yu et al., 2025; Luo et al., 2025c; Team, 2025b; DeepSeek-AI et al., 2025; Yue et al., 2025) recently demonstrates strong reasoning capabilities on math and code tasks, and has emerged as a common practice. These practices utilize verifiers such as Math-Verify (Hynek & Greg, 2025), SandboxFusion (Bytedance-Seed-Foundation-Code-Team et al., 2025), and custom implemented ones (Cui et al., 2025a), which effectively judge the correctness of model rollouts and forgo the need for preference annotations. However, this paradigm is restricted to domains where robust verifiers are available. Moreover, existing implementations of verifiers show inconsistencies (He et al., 2025) since the complexity for rule-based verifiers to handle edge cases is nontrivial. In this work, we propose to extend RLVR practices to domains without robust verifiers.

**Reasoning in General Domains** Previous research explores reasoning in general domains, and a vital part is how to obtain reliable reward signals. One line of work is generative reward models (Mahan et al., 2024), where another generative model judges the quality of rollouts. This concept has been extended to the implementation of verifiers based on a generative model (Ma et al., 2025; Liu et al., 2025a) and enhancements of the judge model itself as a reasoner (Chen et al., 2025). In this work, we demonstrate that reinforcement learning for general-domain reasoning can rely on the decoding probability of the reference answer as a reward. Concurrently, Zhou et al. (2025) utilizes policy likelihood for reference answer as rewards, while limited to short answers less than 7 tokens and requires an auxiliary fine-tuning-based objective. Instead, we find the robustness of per-token probability as a reward signal and extend RLVR to general domains without length constraints.

**Self-Reward Optimization** Unsupervised reinforcement learning on language models using the policy model itself as a reward has recently emerged as an embarrassingly effective approach (Zuo et al., 2025; Zhao et al., 2025). The common idea behind the practice of self-reward is raising the probability of consistent answers (Zuo et al., 2025), intuitively from the observation that concentrating on the majority brings free improvements (Wang et al., 2022). Recent literature (Agarwal et al., 2025) shows that entropy minimization, which naively degrades generation diversity, is a sugar for reasoning tasks, though restricted to certain model families. However, such practice might be problematic for restricting exploration (Cui et al., 2025b; Hochlehnert et al., 2025; Yu et al., 2025). In contrast to self-rewarding methods that remove diversity to exploit existing reasoning ability, our approach builds the reward based on the reference answer, yielding reasoning performance with healthy token entropy from the clip-high trick (Yu et al., 2025).

## 5 CONCLUSION

RLVR shows the power of scaling test-time computation for addressing complex problems and sheds valuable light on paths to AGI. In this work, we present **RLPR**, a novel framework that extends this paradigm to broader general domains. Comprehensive experimental results on Gemma, Llama and Qwen show that our method achieves significant improvement on both general and mathematical reasoning tasks without using external verifiers. We propose a novel probability-based reward and reward debiasing strategy to enhance its quality further. By replacing rule-based reward with probability-based reward, we eliminate the need for external verifiers and achieve better performance than using naive likelihood as a reward or using verifier models. Moreover, we propose a simple standard deviation filtering strategy that stabilizes training by removing samples with low reward standard deviation. In the future, we will explore more domains, including multimodal understanding and scaling **RLPR** to larger models.

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

## A  APPENDIX

### A.1  DISCUSSIONS

**Tokenization.** Our approach tokenizes reference answers for the policy model to calculate probability-based reward scores. A potential concern is the tokenization inconsistency, where the generated token sequence may differ from the token sequence obtained through tokenization, although they can be decoded to the same text. However, our analysis of the RLPR training process on Qwen2.5-7B reveals this is a rare event, occurring in only 0.005% of generated tokens. Furthermore, when these inconsistencies do occur, the language model demonstrates robustness. For example, as shown in Table 3, one sequence might be decoded as tokens ["B", "KD"], while the re-tokenized version becomes ["BK", "D"]. Despite this difference, the model correctly assigns a high probability (i.e., 1.0) to the token "D" following "BK". This suggests that even when faced with unfamiliar token patterns, the LLM can reliably compensate and assign appropriate probabilities.

| Position | Generated Token Sequence | Probability | Tokenized Token Sequence | Probability |
|---|---|---|---|---|
| 1 | DK | 0.5482 | DK | 0.5482 |
| 2 | B | 1.0000 | B | 1.0000 |
| 3 | and | 1.0000 | and | 1.0000 |
| 4 | âĪ | 1.0000 | âĪ | 1.0000 |
| 5 | ␣l | 1.0000 | ␣l | 1.0000 |
| 6 | **B** | **0.9936** | **BK** | **0.0064** |
| 7 | **KD** | **0.9881** | **D** | **1.0000** |
| 8 | are | 0.9141 | are | 0.2629 |
| 9 | the | 0.9655 | the | 0.9585 |
| 10 | same | 1.0000 | same | 1.0000 |
| 11 | angle | 1.0000 | angle | 1.0000 |

Table 3: Token probabilities Comparison between the generated and tokenized token sequences.

### A.2  EXPERIMENTAL DETAILS

Our experiments are conducted on Qwen2.5-7B (Team, 2024) if not additionally specified. Following most RLVR practices, we forgo the supervised fine-tuning process and directly post-train on the base model, and use GRPO algorithm by default. We change the prompt template during training and validation time in our main experiments to control the response structure to have extractable thoughts and answers. The prompt template is shown in Table 4.

---

**RLPR training prompt**

```
<|im_start|>system
A conversation between User and Assistant. The user asks a question,
and the Assistant solves it. The assistant first thinks about the
reasoning process in the mind and then provides the user with the
answer. The reasoning process and answer are enclosed within <think>
</think> and <answer> </answer> tags, respectively, i.e., <think>
reasoning process here </think> <answer> answer here </answer>.
<|im_end|>
<|im_start|>user
{{question}}<|im_end|>
<|im_start|>assistant
```

---

Table 4: We adopt the training prompt of R1 (DeepSeek-AI et al., 2025) for **RLPR**.

#### A.2.1  TRAINING LOGS

We monitor key training metrics of our methods in Figure 6. During training, the response length (Figure 6a) steadily increases, allowing more profound reasoning behaviors and no sign of

| Experiment Name | Table / Figure | Batch Size | Update per Step | Clip Threshold | $\beta$ |
|---|---|---|---|---|---|
| Main Experiment | Figures 1, 6 | 768 | 4 | (0.8, 1.27) | 0.5 |
| | Qwen in Table 1 | 768 | 4 | (0.8, 1.27) | 0.5 |
| | Llama in Table 1 | 256 | 4 | (0.8, 1.27) | 0.9 |
| | Gemma in Table 1 | 256 | 4 | (0.8, 1.27) | 1.0 |
| Ablation Study | Table 2 | 768 | 4 | (0.8, 1.27) | 0.5 |
| Robustness Analysis | Figure 5 | 128 | 1 | (0.8, 1.27) | 0.5 |

Table 5: Implementation setup for each experiment. Default settings align with Sections 3.1 and A.4.

degeneration. In Figure 6b, the policy model quickly learns to follow the response structure. Moreover, as shown in Figure 6c, our training entropy exhibits neither collapses as a result of the clip-high trick, nor abrupt increases. This ensures the balance between exploration and exploitation.

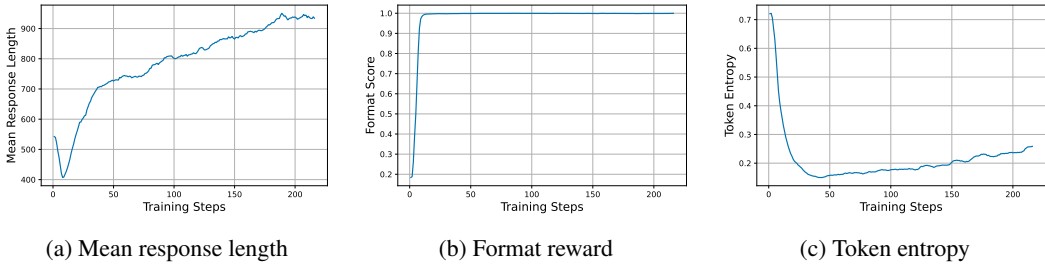

(a) Mean response length  (b) Format reward  (c) Token entropy

Figure 6: Training dynamics of **RLPR** on Qwen2.5-7B

### A.3 TRAINING DATA

We adopt WebInstruct (Ma et al., 2025) as our training dataset, excluding math-related prompts to focus on general-domain reasoning. To ensure the quality and difficulty of training samples, we apply a multi-stage filtering strategy: First, we remove history-related questions and those targeting elementary or middle school levels to avoid commonsense or overly simple content. Finally, leveraging GPT-4.1-mini's reasoning scores (1–4, see Table 6), we retain only highly challenging samples (score $\geq 3$). This process reduces the dataset from 231,833 to 77,687 samples, yielding a focused and high-quality corpus for complex non-mathematical reasoning.

### A.4 IMPLEMENTATION DETAILS

This section provides additional implementation details to supplement Section 3.1. The policy model generates 8 responses per question, using a learning rate of 1e-6. We remove the KL divergence term by setting the KL coefficient to 0. Detailed configurations are presented in Table 5, where the number of policy updates per step and the value of $\beta$ are empirically determined to be optimal for their respective scenarios.

RLVR baselines are trained under the same setting with corresponding **RLPR** results, except using rule-based verifiers and accuracy filtering. For RLVR training on Llama and Gemma, we find accuracy filtering can remove over 90% training prompts and thus significantly increase the training cost and find small batch size causes entropy blow up. So we do not apply accuracy filtering for these two experiments and conduct only one update for each batch to stabilize training.

**Prompt for GPT-4.1 to assess reasoning complexity**

# Description

You are asked to evaluate the reasoning level requirement of problems. Problems are scored from 1 to 4, with higher scores indicating greater reasoning demands. You should make your decision based on the following detail instructions.

## 1 Point: No reasoning requirements.

Problems requiring direct recall of specific facts and commonsense knowledge.
Examples:
- What is Fermat's Last Theorem. (Requires only recalling facts)
- What are the five quantitative forecasting models? (Requires only recalling facts)
- What is the capital of China? (Requires only recalling commonsense knowledge)
- When was Mark Twain born? (Requires only recalling commonsense knowledge)

## 2 Points: No reasoning skill requirements.

Problems that do not require reasoning skills. Either because (1) it is too simply and reasoning skills don't help, or (2) it is too hard to clearly rank different answers since the question is too open-ended and reasoning skills also do not help.

Examples:
- Solve x + 1 = 10, what is the value of x. (Too simple)
- What would you do if you have four legs. (Too open-ended to determine response quality)

## 3 Points: Moderate level reasoning skills and knowledge are enough.

Problems requiring moderate level reasoning skills and knowledge. Such as problems that are mostly likely to be solved by any random undergraduate student regardless of their majors.

Examples:
- Solve a quadratic equation: x² - 5x + 6 = 0.
- Find all solutions to $[\sqrt{x} + 2\sqrt{x^2 + 7x} + \sqrt{x + 7} = 35 - 2x]$. Enter all the solutions, separated by commas.
- Summarize the main causes of World War I. (Requires recalling and organizing established historical factors).
- Describe the importance of empathy in storytelling to someone unfamiliar with the concept, using no more than 4 sentences, and ensure all text is in lowercase. Include a quote from a famous author at the end.

## 4 Points: Long-time analysis and deep understanding of relevant knowledge are required.

Problems requiring long-time to analyze and solve, and depend on deep understanding of relevant knowledge. Such as designing a complex system, developing a comprehensive strategy or providing detail and easy-to-understand solution for realworld problems.

Examples:
- Design a scalable and secure REST API for a large e-commerce platform, considering microservices architecture, data consistency, fault tolerance, and evolving business needs.
- Develop a comprehensive urban planning strategy for sustainable development in a rapidly growing city, integrating environmental, social, economic, and infrastructural considerations.
- Conduct a thorough root cause analysis for a major systemic failure (e.g., a financial crisis or a large-scale environmental disaster) and propose multi-level preventative and corrective policy measures.
- The polynomial $P(x) = (1 + x + x^2 + \ldots + x^{17})^2 - x^{17}$ has 34 complex zeros of the form $z_k = r_k [\cos(2\pi\alpha_k) + i \sin(2\pi\alpha_k)]$, $k = 1, 2, 3, \ldots, 34$, with $0 < \alpha_1 \leq \alpha_2 \leq \alpha_3 \leq \cdots \leq \alpha_{34} < 1$ and $r_k > 0$. Find $\alpha_1 + \alpha_2 + \alpha_3 + \alpha_4 + \alpha_5$.

Please score the following question: Q: {question}
You should first explain your reasoning briefly, then give the final score in following format:
Reasoning score: [1-4]

Table 6: Prompt for GPT-4.1 to assess reasoning complexity.

