# OpenReview forum: "RLPR: Extrapolating RLVR to General Domains without Verifiers"
_ICLR.cc/2026/Conference — Submitted to ICLR 2026_

### Official Review · Reviewer_kKdv · 2025-10-27

**Soundness:** 2
**Presentation:** 3
**Contribution:** 2
**Rating:** 2
**Confidence:** 3

**Summary:**

The paper proposes a "self-reward" strategy for replacing RLVR, using instead the model's own average token log-probability, minus a term for the average token log-probability of the answer without reasoning. A second critical trick is to remove examples for which the standard deviation of this reward is below a running average. In experiments with Qwen2.5, this approach is apparently superior to RLVR trained *with* a verifier, and also apparently superior to a closely-related approach that uses sequence probability instead of average token probability.

**Strengths:**

- The paper tackles an important problem: removing the need for a verifier could in principle make reasoning models applicable in domains in which it is difficult to construct a reasoner.
- The proposed approach is explained clearly.
- The paper's evaluation are relatively comprehensive, including both mathematical and non-mathematical tasks, and going beyond Qwen to include evluations on Llama and Gemma.
- The paper ablates the main methodological components: debiasing, filtering, and per-token probability.

**Weaknesses:**

The idea of the paper is closely related to Verifree, which appeared on arXiv in May. While I could imagine that per-token probability could be superior to sequence probability, it also seems relatively easy to hack, by having reasoning traces that insert a lot of high-probability tokens (like function words).

Given this, the empirical comparison with Verifree is critical, but unfortunately it's hard to know what to make of the evaluations in the two papers. In the Verifree paper, the experiments are performed with Qwen3-8b, and it appears that Verifree improves significantly on Qwen3-8b-base, on MMLU-Pro and SuperGPQA. In this paper, all experiments are performed with Qwen2.5-7b, and the replication of Verifree shows no improvement over Qwen2.5-7B-Inst on MMLU-Pro (no evaluation of SuperGPQA is performed). It's hard to know why Verifree appears to be useless here, and I don't feel confident drawing firm conclusions about sequence vs token reward based on these mixed experiments.

**Questions:**

- Given that the Verifree paper was published in May, would it be possible to conduct comparable experiments on the same datasets and with Qwen3-8B?
- What prevents hacking  the token probability reward by producing reasoning chains that are dominated by high-probability low-content tokens?

---

> ### Author Response · Authors · 2025-12-03
>
> ### Evaluation
>
> We adopt seven reasoning benchmarks covering general domain and math problem solving and argue that including SuperGPQA may only yields marginal information. If this is still required, we could add this results in further revision.
>
> ### Choice of base models.
>
> We conduct experiments on a series of different base models including Qwen2.5, Llama3.1 and Gemma2, and also supplied necessary code for reproduction in the supplementary materials. We agree that it could be possible to retrain our model for every baseline model, but the computation resource could be not affordable for us, so we only reproduces few baselines to save the costs. We believe our comparisons are fair, reliable and reproducible and thus will not be a reasoning for rejection.

---

### Official Review · Reviewer_9ezQ · 2025-10-29

**Soundness:** 3
**Presentation:** 2
**Contribution:** 3
**Rating:** 6
**Confidence:** 3

**Summary:**

This paper proposes RLPR, a verifier-free reinforcement learning framework that uses the model’s own intrinsic probability as a reward signal. Specifically, the method leverages the conditional probability of the ground-truth answer given the model’s CoT output as the reward. This self-probing approach aims to improve performance in domains where external verifiers are unavailable or unreliable. Experiments demonstrate performance improvements across multiple datasets and model families.

**Strengths:**

1. It introduces a new verifier-free RL method leveraging model probability as intrinsic feedback. Overall it is an interesting idea to use the model's own confidence to evaluate the COT and the final answer to assign reward. I can see it is very promising for the field without an easily verifiable answer. The paper also demonstrated empirically the improvements over the general dataset without an easily verifiable answer.

2. The experiments are comprehensive enough to cover both general and math datasets, and the model comparisons are sufficient.

3. Probability Reward formulation is significant enough, and it opens a direction for RL fine-tuning where explicit reward models are hard to define.

**Weaknesses:**

Presentation clarity:
===============
Section 2.2 should have a more clear description. For example, using $z$ and $y$ to denote the response rather than just $o$, then you could use notation $z_1, …, z_n, y_1, …, y_m$. The modified sequence could have more clarity and aligns with previous notations. Otherwise it makes it confusing and might regard the entire sequence as modified. In addition, the $f_seq$ average in l167 should have a more clear definition rather than abusing the notation.

Clarity of the algorithm description:
==========================
1. In Eq.5, it uses $\hat{r}$ directly for the update but in l260 implementation details, the roll out is 8 responses per prompt. It is unclear which algorithm authors are using (PPO or GRPO). Other than that, both algorithms use advantage $A$ to update the policy, the Eq.5 makes it unclear whether the advantage is directly equal to $ \hat{r}$. It is an important detail and needs proper explanations in the paper

2. Which model are you using for the General-Verifier (L342)? It is unclear to me since no model is defined.

3. Figure 3 graphics states o1’, o2’ but captions and texts do not reflect that.

4. Section 3.3 is an important quantitative analysis of the probability reward validity, and it is important to expand it with more details. There should be more data and analysis over the 50 prompts selected in order to understand it better. How many responses for each prompt, percentage for correct and incorrect, how many are math/general data.

Overall the presentation should be improved with more clarity.

**Questions:**

(1) Could the authors report the AUC of  $r’$ (L184 ) to evaluate how well the biased probability aligns with true correctness?

(2) How sensitive is the proposed reward formulation to the base model’s quality? How do other models PR perform in the AUC metric?

(3) Does the method require CoT reasoning to produce reliable probability rewards, or could it also work well with short-form answers?

---

> ### Author Response · Authors · 2025-12-03
>
> ### Usage or r' as reward quality indicator
>
> Since $r'$ is the same for all responses of the same prompt, it might not directly be used in the same way as $r$ to compute the AUC. We adopt such metric in the paper since we think the reward score comparison is meaningful mainly for responses from the same prompt due to the normalization operation of GRPO.
>
> ### Reward sensitivity to basemodel
>
> Thank you for your good question. As we report in Figure 4, even small models like Qwen2.5 0.5B gives high-quality reward surpassing specifically trained general verifier models.
>
> ### Applicability on direct answer
>
> Since we compute the final reward for the reasoning process, so this could not be directly applied to tasks that directly generate final answers after questions are inputed.

---

### Official Review · Reviewer_A8z2 · 2025-10-31

**Soundness:** 2
**Presentation:** 3
**Contribution:** 2
**Rating:** 4
**Confidence:** 4

**Summary:**

This paper introduces RLPR, a verifier-free reinforcement learning method that uses an LLM's intrinsic token probabilities for reference answers as a reward signal to extend RLVR to general domains. Experiments on Qwen2.5-7B, Llama3.1-8B, and Gemma2-2B models show consistent gains on general (e.g., MMLU-Pro, GPQA, TheoremQA) and math benchmarks (e.g., MATH-500, Minerva), outperforming verifier-dependent methods (e.g., General Reasoner) and other verifier-free approaches (e.g., VeriFree).

**Strengths:**

1. This paper addresses an important research question of RL for general domains. The verifier-free approach is a meaningful step toward broadening RLVR to free-form natural language domains, where rule-based or model-based verifiers are impractical or costly.
2. The results are impressive on paper, with RLPR outperforming strong baselines. It also boosts math performance without math-specific data, suggesting transferability.

**Weaknesses:**

1. Using token-level probabilities as a reward signal is conceptually close to likelihood-based or entropy-based self-reward methods (e.g., VeriFree)
2. The paper presents a series of empirical engineering techniques (essentially a "bag of tricks") without providing rigorous theoretical justifications
3. Depending on the LLM's intrinsic probabilities as a reward signal introduces circularity and uncertainty. If the base model is biased, or overconfident, this could amplify errors by reinforcing flawed internal evaluations rather than correcting them. Moreover, the method's generalizability to tasks involving longer, sentence-level answers (e.g., in natural reasoning) is not adequately demonstrated (see question 3)
4. Asserting "sheds valuable light on paths to AGI" in the intro is gratuitous and unsubstantiated

**Questions:**

1. The proposed method seems to have the risk of reward hacking. Have the authors discover any reward hacking in the  trajectories?
2. The debiasing subtracts probs without reasoning. But if the base model already knows the answer, doesn't this penalize good reasoning?
3. Can RLPR generalize to long, compositional, or sentence-level answers, like the answers in natural reasoning?

---

> ### Author Response · Authors · 2025-12-03
>
> ### Reward hacking concern
>
> Our reward relies on ground-truth token labels, so one plausible hacking pattern would be artificially increasing probabilities of frequent textual fragments in the label set. However, throughout training, we did not observe such degenerate behaviors. The model maintains coherent and semantically aligned generation.
>
>
> ### Debias method explanation
>
> For problems which the model already could reliably generate correct answer without explicit reasoning, if adding reasoning cannot further improve the probability reward score, then we categorize this reasoning as unnecessary or low-quality and it receives 0 score. If a good reasoning process further improves the score, then I receives a positive reward score. These behaviors are intended and do not penalize good reasoning.
>
>
> ### Generalize to long output
>
> This is a good question. Our evaluation results on Minerva demonstrates that the model could give competitive results on benchmarks requires complex compositional answers.

---

### Official Review · Reviewer_TbYP · 2025-11-01

**Soundness:** 2
**Presentation:** 3
**Contribution:** 2
**Rating:** 2
**Confidence:** 3

**Summary:**

This paper presents RLPR, a token probability based reward for extending reasoning to non-verifiable domains. RLPR consists of token probability rewards, reward debiasing, and filtering to enhance model performance across multiple domains.

**Strengths:**

- The paper evaluates RLPR across multiple tasks using multiple model families.
- Paper is well-written and easy to follow.

**Weaknesses:**

- Equation 2 seems flawed. Consider two responses "I'm good, not bad" and "I'm bad, not good". Although they are semantically different, Equation 2 seems incapable of distinguishing them in reward. In other words, the token probability reward does not evaluate correctness and is not semantically reliable.
- Why latent factors are additive decomposable? The authors could provide more intuition behind it.
- The choice of Avg@k seems inconsistent. I understand this is due to the budget concern. But showing consist choice of k will make the results more convincing.
- Can the authors also report training dynamics and efficiency using RLPR?
- Sensitivity analysis to parameters such as $\beta$ and temperature is missing.

**Questions:**

See Weaknesses

---

> ### Author Response · Authors · 2025-12-03
>
> ### Equation 2 More Detailed Introduction
>
> **Token orders are not interchangeable, due to the position embedding of LLMs including Qwen, Llama and Gemma.** Equation 2 computes reward over the ordered reference answer sequence $y^\*$, and each reference token $o'_i \in y^\*$ is aligned with its exact decoding probability. Therefore, sequences such as “I’m good, not bad” and “I’m bad, not good” indeed receive different per-token probabilities. This positional dependence is explicitly built into the reward as this is inherently what LLMs commonly do.
>
> ### Decomposability of latent factors
>
> Here we use the notation '+' to mean both factors contributes to the final $U_r$, not conducting an arithmetic operation. If this notations makes it harder to understand, we may update the notation in further revision.
>
> ### Evaluation metrics
>
> We already report both Avg@k and standard error of the mean (SEM) for each benchmark. Not sure which benchmarks contains a too large SEM. If you find certain benchmark results have a large SEM, we may update it to make the evaluation more reliable.
>
> ### Training dynamic
>
> We also already shown the training reward, response length and entropy in Figure 5 and also format reward detail in appendix. We may supply wandb log in further revision if needed.
>
> ### Parameter sensitivity of beta
>
> This is a good question, we found beta could be sensitive but we could easily find a good score by examine the reward of remained samples  by conduct the training for only few steps. We did not conduct comprehensive full training comparison due to resource limitation.

---

### Meta-Review · Area_Chair_P7gd · 2026-01-04

**Summary:**

This paper studies the use of answer token probabilities as rewards in reinforcement learning fine-tuning of language models. The main points made by the reviewers are the following.

Strengths:
1. Good/clear writing (TbYP, kKdv)
2. Evaluation across many tasks/models (TbYP, A8z2, 9ezQ, kKdv)
3. Method found interesting and potentially useful (A8z2, 9ezQ, kKdv)

Weaknesses/concerns:
1. Grossly incorrect presentation, including mathematical notation -- symbols not being defined, etc. (TbYP, 9ezQ). I strongly agree with this concern; it is latently behind TbYP's first weakness as well: in equation (2) $f_{seq}$ seems to take a set as input instead of a sequence, $n$ and $N'$ are undefined, etc.
2. Intuitions behind some of the modelling and evaluation choices not well explained (TbYP, A8z2)
3. Questions about analysis of training dynamics, efficiency, base model, various other modelling choices, and generalisation to shorter/longer reasoning chains (TbYP, 9ezQ, kKdv)
4. Limited conceptual novelty, particularly relative to Verifree (A8z2, kKdv)
5. Concerns about accumulation of bias and reward hacking (A8z2, kKdv)

Considering the lack of a thorough response and the importance of the unaddressed weaknesses -- including correct mathematical presentation -- I recommend rejection.

**Reviewer Concerns:**

The authors' responses were quite short and many of the comments were unaddressed.
1. Not addressed in the response because not explicitly stated as a weakness, but only some of the resulting questions from the reviewers (TbYP, but not 9ezQ) were answered. This point remains an important weakness of the paper.
2. Not addressed with sufficient detail in the response; hand-waving explanations were given to some points.
3. Some speculative answers given; a few existing experiments were pointed to in the response, and some new ones promised but not provided. Overall, the questions about missing analyses remain largely unaddressed, though they are important for understanding why the method works and its possible failure cases.
4. Entirely unaddressed in the response.
5. Addressed in the answer to A8z2, but not to kKdv.

**Reviewer Scores:**

The original scores were 2, 4, 6, 2. Given that many of the important concerns raised by the reviewers were not adequately addressed, it is likely that all reviewers would keep their original scores.

Note that even with the two most negative reviewers (2) increasing their scores to 4, the average would fall below borderline.

---

### Decision · Program_Chairs · 2026-01-26

Reject